# Homologous Drought-Induced 19 Proteins, PtDi19-2 and PtDi19-7, Enhance Drought Tolerance in Transgenic Plants

**DOI:** 10.3390/ijms23063371

**Published:** 2022-03-21

**Authors:** Caijuan Wu, Miao Lin, Feng Chen, Jun Chen, Shifan Liu, Hanwei Yan, Yan Xiang

**Affiliations:** 1Laboratory of Modern Biotechnology, School of Forestry and Landscape Architecture, Anhui Agricultural University, Hefei 230061, China; tracey0920@sina.com (C.W.); linmiao199810@163.com (M.L.); cfahau357@sina.com (F.C.); chenjun19940819@sina.com (J.C.); liushifan0321@163.com (S.L.); hwyanahau@163.com (H.Y.); 2National Engineering Laboratory of Crop Stress Resistance Breeding, College of Life Sciences, Anhui Agricultural University, Hefei 230061, China

**Keywords:** *PtDi19-2*, *PtDi19-7*, drought stress, ABA, stomatal closure, lateral bud dormancy

## Abstract

Drought-induced 19 (Di19) proteins play important roles in abiotic stress responses. Thus far, there are no reports about Di19 family in woody plants. Here, eight Di19 genes were identified in poplar. We analyzed phylogenetic tree, conserved protein domain, and gene structure of Di19 gene members in seven species. The results showed the Di19 gene family was very conservative in both dicotyledonous and monocotyledonous forms. On the basis of transcriptome data, the expression patterns of Di19s in poplar under abiotic stress and ABA treatment were further studied. Subsequently, homologous genes *PtDi19-2* and *PtDi19-7* with strong response to drought stress were identified. PtDi19-2 functions as a nuclear transcriptional activator with a transactivation domain at the C-terminus. PtDi19-7 is a nuclear and membrane localization protein. Additionally, PtDi19-2 and PtDi19-7 were able to interact with each other in yeast two-hybrid system. Overexpression of *PtDi19-2* and *PtDi19-7* in *Arabidopsis* was found. Phenotype identification and physiological parameter analysis showed that transgenic *Arabidopsis* increased ABA sensitivity and drought tolerance. *PtDi19*-7 was overexpressed in hybrid poplar 84K (*Populus alba × Populus glandulosa*). Under drought treatment, the phenotype and physiological parameters of transgenic poplar were consistent with those of transgenic *Arabidopsis*. In addition, exogenous ABA treatment induced lateral bud dormancy of transgenic poplar and stomatal closure of transgenic *Arabidopsis*. The expression of ABA/drought-related marker genes was upregulated under drought treatment. These results indicated that *PtDi19-2* and *PtDi19-7* might play a similar role in improving the drought tolerance of transgenic plants through ABA-dependent signaling pathways.

## 1. Introduction

Drought stress is one of the most common threats to plants. Drought stress leads to the accumulation of a large number of harmful substances, causing serious damage to plant cell structure, resulting in the reduction of plant biomass or even normal growth [1]. To adapt to the damage caused by drought stress, plants hold highly evolved mechanisms in the terms of phenotype and physiology, such as increasing the number and length of roots to improve water absorption efficiency [2], adjusting the degree of stomata opening [3], and increasing the surface area of the leaves to avoid water loss [4]. In addition, at the molecular level, genes respond to drought stress by activating signal transduction pathways, resulting in physiological and biochemical responses. Long-term studies have demonstrated that the endogenous hormone abscisic acid (ABA) is closely related to the regulation of drought resistance in plants [5,6].

Transcription factors (TFs) play a vital role in plant response to abiotic stress. Zinc-finger protein is the most prominent transcription regulator in plants. On the basis of the number and order of the Cys and His residues that bind the Zinc ion in the secondary structure of the finger, they were divided into several different types, including C2H2, C2C2, C2HC, C2C2C2C2, and C2HCC2C2 [7,8,9]. Among the different zinc-finger proteins, Cys2/His2 (C2H2)-type (TFIII-types) was a characteristic representative of eukaryotic transcription factors to abiotic stress [10,11,12,13,14,15]. Di19 (drought-induced 19) protein containing two typical Cys2/His2 zinc finger domains is one of the Cys2/His2-type TFs.

Di19 protein was first identified in *Arabidopsis thaliana* in 2006, and subsequently studies have demonstrated that the members play the roles in growth and development as well as coping with abiotic stresses in higher plants [16,17,18,19,20]. There are seven Di19 genes in rice—*OsDi19-4* can not only interact with its own family members to participate in drought tolerance, but also interact with *OsCDPK14* to form protein heterodimers [21]. *OsCDPK14* phosphorylated *OsDi19-4*, and the phosphorylation was further enhanced by ABA treatment [22]. In cotton and soybeans, Di19s were involved in regulating plants tolerance to high salt and ABA treatment [23,24]. Another study showed that transgenic *Arabidopsis* of wheat gene *TaDi19A* was more sensitive to salt, ABA, and mannitol, especially root elongation to salt stress [25]. Recently, Di19 protein has been assigned to be involved in hormonal interactions in plant development. *AtDi19-3* interacted with *AtIAA14* as a positive regulator of auxin signaling and plays a role in some ethylene-mediated responses in *Arabidopsis* [26]. These studies suggested that Di19 family genes play crucial role in growth and development and abiotic stress response.

Poplar (*Populus* sp.) is a typical representative of woody plants. It is mainly distributed in north and southwest China (most of these areas are arid areas) [27]. As an excellent industrial tree species, it is an important source of pulp, wood, and biofuels, and has important ecological and economic value [28]. Currently, poplars face the most severe environmental drought stress, causing water loss, stomatal closure, and osmoregulation imbalances [29,30,31]. Therefore, it is necessary to study drought resistance genes, excavate their regulation mechanism, and breed drought-tolerant poplar varieties. Drought-induced 19 family genes, as a class of important transcription factors, and their functions in poplar are unclear.

Here, we identified eight Di19 genes in the poplar. Subsequently, we provide detailed information, including phylogenetic analysis, gene structure, and conserved protein domain of Di19s in seven species. The promoter analysis and syntenic relationships of PtDi19s were investigated. The expression pattern of PtDi19 was analyzed by RNA-seq data analysis and quantitative real-time PCR (qRT-PCR) under different stresses, and stress-related homologous genes, *PtDi19-2* and *PtDi19-7*, were isolated. We obtained overexpression of *PtDi19-2* and *PtDi19-7* in *Arabidopsis* and overexpression of *PtDi19-7* in 84K poplar. The drought resistance and ABA sensitivity of overexpressed plants were investigated. The results suggested that *PtDi19-2* and *PtDi19-7* were involved in the positive regulation of ABA-dependent signaling pathways and drought tolerance. The findings of this study enrich our understanding of PtDi19 gene family and will contribute to the genetic improvement of woody plants drought tolerance.

## 2. Results

### 2.1. Identification and Bioinformatics Analysis of the PtDi19 Genes

Eight PtDi19 genes were identified and distributed randomly on 8 of the 19 chromosomes in poplar. We named *PtDi19-1*-*PtDi19-8* according to their position on chromosomes. The basic characteristics of the eight PtDi19 members, such as their protein length, pI, and MW, are listed in Appendix A.

We analyzed synteny relationship of PtDi19 genes in the *P. trichocarpa* genome (Appendix A). Eight PtDi19 genes were distributed randomly on different chromosomes. In addition, except for gene *PtDi19-5*, all genes have homologs. The results showed that part of the PtDi19 genes may be generated by whole genome duplication (WGD), suggesting that WGD event was the main driving force of PtDi19 evolution. To understand the evolutionary constraints of the PtDi19 family, we calculated the nonsynonymous to synonymous substitution ratios (Ka/Ks) of PtDi19 gene pairs (Appendix A). The Ka/Ks ratio of 12 homologous genes was less than 1, which indicated that the PtDi19 gene family of poplar has experienced a strong purification selection pressure during its evolution.

Some studies have demonstrated that genes with similar promoter homeopathic elements often have similar expression patterns [32]. To investigate the regulatory mechanism of PtDi19 family genes, the PlantCARE database was used to process and analyze the 2 kb upstream sequence of PtDi19 family genes. Various cis-acting elements were detected in the promoter region of PtDi19 genes, including stress response elements, developmental response elements, and hormone response elements, suggesting that expression of PtDi19s was regulated by a complex network (Appendix A).

### 2.2. Analysis of Evolutionary Trees, Conserved Protein Domain, motifs, and Gene Structure

To study the evolutionary relationship of Di19 gene family in seven species, including eight in poplar, seven in *Arabidopsis thaliana*, seven in rice, seven in soybean, three in papaya, two in grape, and two in cotton, the Di19 proteins were divided into three subfamilies (S1–S3) on the phylogenetic tree (Figure 1A). Among them, there are 24 members in S1 subfamily, including six members in poplar (75%), five in *Arabidopsis* (71%), five in soybean (71%), two in cotton (100%), two in rice (28%), one in grape (50%), and one in papaya (33%). Still, *PtDi19-6* and *PtDi19-5* were distributed in S2 and S3, respectively. Taken together, these results showed that most PtDi19 proteins were phylogenetically closer to AtDi19 proteins, while *PtDi19-5* and *PtDi19-6* proteins were more closely related to rice Di19 proteins (Figure 1A). We used the CDD tool in NCBI to identify two conserved domains (zf-Di19 and Di19_C) in the 36 Di19 protein sequence [33]. It is worth noting that all Di19 proteins have both structures, which may indicate that they have similar biological functions (Figure 1B). Meanwhile, MEME was used to further analyze the conserved motifs of Di19 family. Among 36 Di19 genes, 20 conserved motifs were identified as motifs 1 to 20 (Appendix A). In 36 Di19 protein sequences, motif 1 accounted for 100%, and motif 2, motif 3, motif 4, motif 5, and motif 6 accounted for about 70% of the total genes (Appendix A). The results indicated that the six motifs might have important components on Di19s. The gene structure analysis showed that two Di19 genes contained three introns (5.5%), 31 genes contained four introns (86%), two genes contained five introns (5.5%), only *AtDi19-4* contained six introns (Figure 2).

### 2.3. Expression Profiles of PtDi19 Genes under Different Stresses

We mapped the expression profiles of eight PtDi19 genes in different tissues under drought, salt, and cold through the published transcriptome dataset (Figure 3A). According to the heat map, most PtDi19 genes in leaves were induced by prolonged drought treatment; *PtDi19-7* was especially significantly up-regulated, followed by *PtDi19-2*. Because *PtDi19-7* and *PtDi19-2* also responded strongly to salt and cold, the two genes were clustered into a group on the heat map. The performance of other genes in the heat map was not obvious.

According to previous studies in other plants, the Di19 genes were involved in drought, salt, and temperature stress responses, and might depend on the ABA signaling pathway. To verify whether the members of the PtDi19 gene were affected by abiotic stress and hormone treatment, we analyzed the relative expression of Di19 genes in poplar under PEG, NaCl, ABA, and cold stress by qRT-PCR. The results showed that after PEG treatment, except for *PtDi19-1* and *PtDi19-6*, the relative expression levels of the other six PtDi19s decreased first and immediately increased after 1 h and reached the highest level at 6 or 12 h, of which *PtDi19-2* and *PtDi19-7* induced about 10 times that of the control (Figure 3B). After NaCl treatment, apart from *PtDi19-8*, the relative expression of all genes increased. The expression of *PtDi19-1* and *PtDi19-2* were upregulated approximately fourfold, and even *PtDi19-6* increased by about sevenfold (Figure 3C). The expression of *PtDi19-4* and *PtDi19-7* was markedly induced by ABA. *PtDi19-1*, *PtDi19-3*, *PtDi19-6,* and *PtDi19-8* performed the same expression pattern, decreased at 3 h, increased at 6 h and 12 h, and finally decreased slightly (Figure 4A). Overall, eight PtDi19 genes were observably induced by cold treatment. The relative expression level of *PtDi19-7* reached the highest peak at 12h, which was more than 30 times of the control. *PtDi19-3* was induced about 10 times that of the control (Figure 4B). The above results showed that the Di19 gene in poplar had different degrees of response to abiotic stress and might have great potential in abiotic stress. Combined with transcriptome and qRT-PCR analysis, *PtDi19-2* and *PtDi19-7* responded significantly to drought, PEG, and ABA treatment. Thus, we selected them as candidate genes for subsequent experimental studies.

### 2.4. Subcellular Localization and Transcriptional Activity

It was reported earlier that the Di19 transcription factor GFP signal was located in the nucleus or in the nucleus and membrane. *PtDi19-2* and *PtDi19-7* were cloned to construct 1305-PtDi19-2 and 1305-PtDi19-7 vectors, and 35S-GFP was used as the positive control group. The results showed that PtDi19-2 was located in the nucleus of tobacco, verified by 4’,6-diamidino-2-phenylindole (DAPI) staining (Figure 5A). PtDi19-7 was a nuclear membrane localization protein (Figure 5B).

In order to further study whether PtDi19-2 and PtDi19-7 had transcriptional activity, we chose pGBKT7-53+pGADT7-T as the positive control group; pGBKT7-PtDi19-2, pGBKT7-PtDi19-N-C, pGBKT7-PtDi19-N, and pGBKT7-PtDi19-7 as the experimental group; and pGBKT7 empty as the negative control group (Figure 6A). The strains were showed growth on SD/-Trp and SD/-Ade/-His/-Trp/X-α-Gal plates. PGBKT7-53 + PGADT7-T yeast cells were able to grow normally and turn blue (Figure 6B). The negative control and pGBKT7-PtDi19-2-N yeast cells were unable to turn blue, but pGBKT7-PtDi19-2, pGBKT7-PtDi19-2-N-C, and pGBKT7-PtDi19-7 yeast cells were able to grow normally on the SD/-Ade/-His/-Trp/X-α-Gal plate and turn blue. The results showed that PtDi19-2, PtDi19-N-C, and PtDi19-7 had transcriptional activity. The PtDi19-2-N had no transcriptional activity. Therefore, the transactivation activity of PtDi19-2 protein might exist in the C-terminal domain.

### 2.5. PtDi19-2 Interactions with PtDi19-7 as Co-transcription Factors

In order to verify the interaction between PtDi19-2 and PtDi19-7, we performed yeast two-hybrid experiment. As shown in Figure 6C, BD-PtDi19-2-N and AD-PtDi19-7 co-transformed yeast cells grew well and turned blue on SD/-Leu/-Trp/-His/-Ade/X-α-Gal selective medium. With the continuous dilution of bacterial solution concentration, the experimental group was still able to turn blue. The finding suggested that PtDi19-2-N interact with PtDi19-7 in yeast.

### 2.6. Transgenic Arabidopsis Improved Sensitivity to ABA and Enhanced Drought Tolerance during the Germination Period

We overexpressed *PtDi19-2* and *PtDi19-7* in *Arabidopsis thaliana* to study their biological functions in plants. Through standardized culture and strict positive screening, the third-generation seeds of *Arabidopsis* transgenic lines (OE2-3, OE2-4, OE7-5, and OE7-9) were obtained (Appendix A). Firstly, we evaluated the response of these transgenic lines to stress during the seeding stage. On 1/2MS solid medium, there was no difference in seed germination between WT and transgenic lines. The germination of transgenic lines was significantly inhibited under 0.7 µM ABA treatment. In addition, some WT seeds were able to grow normally, while the transgenic lines only germinated and almost did not develop roots (Figure 7A). The results showed that the *PtDi19-2* and *PtDi19-7* increased the sensitivity of transgenic plants to ABA. On 1/2 MS solid media with different concentrations of mannitol (300, 350, and 400 mM), it could be seen that with increasing mannitol concentration, the growth of both WT and transgenic lines were inhibited, and the damage of mannitol on transgenic lines was less than WT (Figure 7A). The germination rate of WT decreased by 84.25%, and the germination rate of OE2-3, OE2-4, OE7-5, and OE7-9 decreased by 64.41%, 61.67%, 57.93%, and 61.11%, respectively (Figure 7B). These results indicated that the *PtDi19-2*-overexpressing and *PtDi19-7*-overexpressing lines improved the drought tolerance of transgenic *Arabidopsis* during the germination period.

### 2.7. Overexpression of PtDi19-2 and PtDi19-7 Could Enhance Drought Tolerance of Transgenic Arabidopsis

Transgenic lines showed resistance to drought during the germination stage. What is the function of transgenic plants at seedling stage? We stopped water supplementation after three weeks of growth of WT, OE2-3, OE2-4, OE7-5, and OE7-9 until significantly different phenotypes appeared. Under normal growth conditions, WT and overexpressed lines showed no significant difference and grew well. After 10 days of drought treatment, WT showed obvious wilting compared with transgenic lines. The growth inhibition of transgenic lines was dramatically reduced after 3 days of rehydration (Figure 8A). The survival rate of the transgenic lines were 100%, while the survival rate of WT was only 16.6%.

Under drought treatment, leaves in the transgenic lines showed significantly higher relative water content than those of WT, showing that transgenic lines have good water retention capacity (Figure 8B). In addition, the electrolyte leakage of *PtDi19-2*-overexpressing and *PtDi19-7*-overexpressing lines was lower than that of wild-type plants after drought stress (Figure 8C). Proline played a key role in regulating intracellular osmotic potential, helping subcellular stability and protecting plants from osmotic damage under drought stress [34]. The proline content of *PtDi19-2* and *PtDi19-7* transgenic plants was apparently higher than that of WT (Figure 8D). POD could remove superoxide ions and hydrogen peroxide [35]. The POD activity measurement showed that POD activity of each line increased after drought treatment, yet overexpression lines were higher than that of the wild type (Figure 8E). Moreover, the contents of MDA were important reference indicators for plant drought resistance. MDA was related to the degree of membrane lipid peroxidation [36]. The research showed that the MDA content of all lines in the control group was in a relatively low and stable state in the normal growth condition. After drought treatment, MDA content of transgenic plants decreased compared with wild type (Figure 8F). DAB staining showed that the overexpression lines had less accumulation of H_2_O_2_ in the plants compared with WT (Figure 8G). Therefore, *PtDi19-2*-overexpressing and *PtDi19-7*-overexpressing lines improved the drought resistance of plants by reducing cell damage under drought stress.

### 2.8. PtDi19-7 Overexpression Could Enhance Drought Tolerance of Transgenic 84K Poplar

To further explore the function of *PtDi19-7* in poplar, we obtained 84K poplar transgenic lines of *PtDi19-7* through poplar genetic transformation, and the positive lines of *PtDi19-7* were identified by CUS staining and PCR (Appendix A). On day 0, there were no significant differences in phenotype and physiological parameters between *oxPtDi19-7* and wild-type 84K poplar (Figure 9A). On the eighth day of drought treatment, the terminal bud of wild-type 84K Poplar withered, and other leaves also showed different degrees of water loss. However, the overexpressed lines showed no signs of water loss and grew well. After rehydration, it could be seen that most leaves of wild-type 84K poplar had died, and petiole could not be directly extended. The results of physiological parameters showed that the leaf relative water content of the overexpressed lines was significantly higher than that of wild-type 84K poplar (Figure 9B). Chlorophyll content in plant leaves can also reflect the tolerance of plant to abiotic stress to a certain extent. The results showed that the chlorophyll content of *oxPtDi19-7* was higher than that of wild-type 84K poplar after drought treatment (Figure 9D). In addition, under drought treatment, MDA content and relative electrolytic leakage of *oxPtDi19-7* were lower than those of wild-type 84K poplar (Figure 9E,G), and proline content and POD activities were higher than those of wild-type 84K poplar (Figure 9C,F). These results indicated that overexpression of *PtDi19-7* could improve drought tolerance of 84K poplar. It is worth noting that the above results were consistent with those of *Arabidopsis*.

The root system is an important functional organ of forest absorbing and transferring soil resources. The formation of deep roots can provide a way for poplars to absorb deep soil water resources, which is a key functional trait for poplars to cope with drought environment. After eight days of drought treatment, the root system of each line was taken out to measure their length, fresh weight, and dry weight (Figure 10A). The results showed that the root length of *oxPtDi19-7* lines were significantly longer than that of wild-type 84K poplar (Figure 10B), and the dry weight and fresh weight of overexpressed lines were all higher than that of wild-type 84K poplar (Figure 10C,D). This might be important evidence that *PtDi19**-7* improves the drought tolerance of transgenic lines.

### 2.9. ABA-Induced Lateral Bud Dormancy in 84K Poplar

*PtDi19-7* were sensitive to ABA in *Arabidopsis*. To reveal the effect of ABA on *oxPtDi19-7*, we conducted an ABA-induced lateral bud dormancy experiment. The results showed that lateral bud growth of *oxPtDi19-7* lines was inhibited on 1/2 MS medium supplemented with 5 µM ABA (Figure 10E). Lateral bud growth of *oxPtDi19-7* lines were dormant on 1/2 MS medium supplemented with 10 µM ABA. These results indicated that *oxPtDi19-7* was responsive to ABA, which was consistent with the results of *Arabidopsis* overexpressed lines.

### 2.10. Overexpression of PtDi19-2 and PtDi19-7 Promoted ABA-Induced Stomatal Closure and Upregulated Expression of Stress-Response Genes

Many studies have revealed that stomatal opening is affected by ABA, and leaf water loss is related to stomatal adjustment [37]. We examined the regulation effect of *PtDi19-2* and *PtDi19-7* overexpression *Arabidopsis* on stomatal opening in plants by ABA. Changes in stomatal size of WT and overexpressed plants without treatment or 1 µM ABA treatment were detected. The observation results of electron microscopy showed that stomata in all samples were open, partially open, and closed (Appendix A). Under normal circumstances, there was no significant difference in the opening degree of the three kinds of stomata in plants (Appendix A). Under 1 µM ABA treatment, most stomata of transgenic lines changed from fully open and partially open to completely closed. These results indicated that *PtDi19-2* and *PtDi19-7* might play a vital role in ABA-mediated stomatal closure.

To clarify the role of *PtDi19-2* and *PtDi19-7* in the regulation of drought stress tolerance, we analyzed the expression of four drought/ABA stress genes (*AtABF3*, *AtDREB2A*, *AtERD1*, and *AtRD29A*) in drought treatment [38,39,40,41]. The results showed that the four drought/ABA stress genes were upregulated after 10 days of drought stress by qRT-PCR (Appendix A–D). Among them, *AtABF3* had the highest expression level, which had increased by about 10 times compared with wild-type plants. Therefore, the increased drought resistance of *PtDi19-2* and *PtDi19-7* overexpression lines might be due to the regulation of drought related gene expression by ABA signaling pathway.

## 3. Discussion

Di19 protein is a type of zinc finger protein, which belongs to Cys2/His2 (C2H2) zinc-finger protein. Zinc finger protein has strong function and has outstanding performance in coping with abiotic stress [42]. The first Di19 protein was found in *Arabidopsis thaliana* [19]. *AtDi19-1* and *AtDi19-3* were significantly upregulated under drought stress. *AtDi19-2* and *AtDi19-4* showed strong induction under high salt treatment [19]. The function of *AtDi19-7* differs from other AtDi19s, being involved in regulating light signal [20]. *AtDi19-1* was involved in drought stress response in *Arabidopsis thaliana* by binding to pathogenic promoters (*PR1*, *PR2*, and *PR5*) [16]. In addition, Di19 protein was discovered in rice, moso bamboo, cotton, soybean, and other species, all of which are involved in plant stress response [17,22,24,43]. However, the function of Di19 in the woody plant poplar has not been characterized. In this study, we identified eight highly conserved Di19 genes in poplar. Previous studies demonstrated that most species have few Di19 family members, such as *Arabidopsis* (7), cotton (2), and rice (7). We compared the Di19 proteins of seven species and analyzed their evolutionary relationship. Through the analysis of phylogenetic tree, conserved protein domain, and gene structure, it was discovered that most Di19s have two typical Cys2/His2 zinc-finger domains (Figure 1B). Intron was an important component of plant gene structure and had a variety of functions. The deletion or change of intron will lead to structural changes and affect the evolution of gene family [44]. In this study, nearly all Di9 genes contained introns mainly located in two Cys2/His2 zinc-finger domains, forming four distinct intron patterns. The most common structural pattern contained five exons and four introns (Figure 2). Notably, all eight PtDi19 had only five exons. The above results meant that the evolutionary process of Di19 transcription factor was conservative rather than accidental mutation.

Studies have demonstrated that exploring cis-elements in the upstream region of a gene can help predict the transcriptional regulation of the gene family [45,46]. Our analysis showed that the promoter region of the PtDi19 gene contains various cis-acting elements (Appendix A). It is worth noting that the promoter region of *PtDi19-2/7* gene was rich in MYB and MYC cis-acting elements (Appendix A). MYB elements were found on promoters of many anti-stress genes. MYB transcription factors exist widely in plants, and they combine with MYB elements and participate in the regulation of plant response to the external environment, especially stress [47]. MYC element is a cis-acting element in response to drought and ABA. The core sequence is CANNTG, which is present on the promoters of various stress resistance genes [48]. In maize, the promoter region of *ZmDi19-1* contains both MYB and MYC elements, and its expression can be induced by ABA, PEG, and sodium chloride stress [49]. Furthermore, most Di19 members of soybean contained MYB elements, and the transcription level of soybean Di19 members was higher under salt, drought, oxidation, and ABA stress [24]. On the basis of the RNA-seq data of poplars, we analyzed the expression patterns of eight PtDi19s in different tissues under six stress treatments. The results showed that some Di19 genes in poplars were significantly induced by drought (*PtDi19-2/7*) (Figure 3A). *PtDi19-7* expression level was the highest in leaves treated with prolonged drought, and its homologous gene *PtDi19-2* was also induced under prolonged drought treatment. The qRT-PCR experiment also showed a similar trend. The *PtDi19-7* had a higher expression level under PEG and ABA treatments, which reflected that *PtDi19-7* transcription level was higher under drought stress and hormone treatment, and *PtDi19-2* showed the same trend (Figure 3B,D). Therefore, MYB and MYC cis-acting element might be the reason behind the stress-regulated expression of *PtDi19-2/7*.

Most Di19 proteins are located in the nucleus, such as AtDi19-1, AtDi19-7, GhDi19-1, GhDi19-2, GmDi19-5, and OsDi19-4 [17,19]. Some are located in the nucleus and cell membranes, such as OsDi19-7, ZmDi19-1, and TaDi19A [25,49]. In our study, PtDi19-2 was found to be located in the nucleus (Figure 5A), and PtDi19-7 was located in the nucleus and cell membrane (Figure 5B). The results also showed the potential functional diversity of PtDi19 gene. *Arabidopsis* Di19-3 and maize Di19-1 showed transcriptional activation activity in yeast. Our investigation showed that PtDi19-2 and PtDi19-7 both act as transcriptional activators in yeast cells (Figure 6B). In rice, the truncated C-terminal of OsDi19s (except OsDi19-4) did not show any transcriptional activity in yeast [22]. This feature also existed in poplar trees. After truncating the C-terminal fragment of PtDi19-2, PtDi19-2-N had no transcriptional activation activity (Figure 6B). This suggested that the C-terminal might be necessary for PtDi19-2 transcriptional activation. Interestingly, our research found that the truncated PtDi19-2-N without transcriptional activation could interact with PtDi19-7 in yeast (Figure 6C). These results suggested that PtDi19-2 and PtDi19-7 were transcription factors and interacted with each other in the yeast two-hybrid system.

To further study the role of poplar Di19s in plants under abiotic stress, we obtained overexpression of *PtDi19-2* and *PtDi19-7* in *Arabidopsis*. The germination test showed that the overexpression lines of *PtDi19-2* and *PtDi19-7* had higher resistance to mannitol and sensitivity to ABA than the wild type (Figure 7). Some studies showed that the suppression of a negative regulator or the enhancement of a positive regulator of ABA appears to confer drought tolerance [50,51,52]. In addition, exogenous ABA inhibited lateral bud development of *oxPtDi19-7* (Figure 10E). In *Arabidopsis*, the stomata of the overexpression lines were essentially closed after the seedlings were treated with 1 µM ABA (Appendix A). Studies have shown that ABA could maintain seed dormancy, prevent seed germination, and induce stomatal closure [53]. Therefore, we inferred that the response of *PtDi19-2* and *PtDi19-7* to drought stress was ABA-dependent. Overall, we supposed that *PtDi19-2* and *PtDi19-7* play a positive regulatory factor in ABA and drought response.

Seedlings were exposed to natural drought for 10 days, and the transgenic plants showed stronger drought tolerance (Figure 8A). The further analysis showed that overexpression of *PtDi19-2* and *PtDi19-7* have higher relative water content (Figure 8B), proline content (Figure 8D), and POD activity (Figure 8E); lower electrolyte leakage (Figure 8C) and MDA content (Figure 8F); and less hydrogen peroxide accumulation (Figure 8G), using WT as a control. In addition, *oxPtDi19-7* also showed a drought resistance phenotype, and the physiological parameters showed the same trend as *Arabidopsis* (Figure 9). These results indicated that *PtDi19-2/7* play a crucial role in abiotic stress tolerance. To investigate whether *PtDi19-2* and *PtDi19-7* affect the expression of genes related to drought stress and ABA signaling, some marker genes were analyzed in *Arabidopsis*. The results showed that the expression levels of *AtABF3*, *AtDREB2A*, *AtERD1*, and *AtRD29A* in transgenic lines were significantly changed (Appendix A–D). A previous study indicated that the *ABF3*, a bZIP transcription factor, could positively regulate ABA signal transduction [54]. Therefore, *PtDi19-2* and *PtDi19-7* act as positive regulators of poplar response to drought stress and ABA pathway.

## 4. Materials and Methods

### 4.1. Identification and Bioinformatics Analysis of Di19 Gene Family

The Di19 proteins, annotated the presence of domains of zf-Di19 (PF14571) and Di19_C (PF05605), and the chromosomal positions were selected from the annotation file of poplar genome (https://phytozome.jgi.doe.gov/pz/portal.htmL; accessed on 18 January 2022) [55]. Molecular weight (MW) and isoelectric point (pI) of poplar Di19 genes was obtained from ExPASy website (http://www.expasy.org/tools/; accessed on 18 January 2022). Di19 gene sequences of other species, including *Arabidopsis*, soybean, rice, cotton, papaya, and grape, were collected from the phytozome database.

We put the *P. trichocarpa* genome file and gff3 file into One Step MCScanX in TBtools, and set the execution standard match size: 5 and E-value: 1E-05 to study collinearity relationships between PtDi19 members [56]. The non-synonymous substitution rate (Ka) and the synonymous substitution rate (Ks) of these homologous gene pairs were calculated with the simple Ka/Ks calculator in TBtools (https://github.com/CJ-Chen/TBtools; accessed on 28 January 2022) and presented using a three-line table [57]. The divergence time is estimated by the formula T = Ks/2r (r = 1.5 × 10^−8^) [58]. To understand the cis-acting elements of PtDi19 gene family, the upstream sequences (2000 bp) of the translation start site (TSS) of eight PtDi19s were extracted from the poplar genomic sequences, and then were submitted to PlantCARE website (http://bioinformatics.psb.ugent.be/webtools/plantcare/html/; accessed on 28 January 2022) [59].

### 4.2. Phylogenetic Tree, Gene Structure, and Conserved Motif Analysis

On the basis of the multiple sequence alignment of Di19s full-length protein sequences of seven species, a phylogenetic tree was constructed using MEGA 6.0 software by the neighbor-joining method to explore the evolutionary relationships on Di19s [60]. The Gene Structure Display Server (GSDS: http://GSDS.cbi.pku.edu.cn/; accessed on 28 January 2022) was used to construct a schematic diagram of the Di19 gene structure [61]. The domain detection of 36 Di19 protein was performed using the Web CD-Search Tool in NCBI (https://www.ncbi.nlm.nih.gov/Structure/bwrpsb/bwrpsb.cgi; accessed on 28 January 2022) [33]. In addition, we submitted the protein sequences to MEME website (http://MEME-suite.org/; accessed on 28 January 2022) to elucidate the conserved motif structure of Di19 genes in seven species.

### 4.3. Transcriptome Data to Analyze the Expression Patterns of PtDi19

The published RNA-Seq data on the different tissues and stages of abiotic tolerance was downloaded from E-MTAB5540 [62]. The log_2_ (FPKM+1) value for each Di19 family gene was used to generate heat maps using TBtools software.

### 4.4. Plant Materials, Treatments, and RNA Extraction

The seedlings of *Populus deltoides cv. ‘Nanlin 95’* were grown for one and a half months in a tissue culture laboratory (14 h light from 08:00 to 22:00) at 25–28 °C. Then, the plants were treated independently with abiotic stress treatment, including 20% PEG-6000 solution, 200 mM NaCl solution, 100 µM ABA, and cold treatment [63]. Meanwhile, the seedlings treated with water were used as control group. The experimental group and control group were sampled at 0 h, 1 h, 3 h, 6 h, 12 h, and 24 h. The samples were immediately frozen with liquid nitrogen and stored at −80 ℃. RNA extraction of all samples in the text was using the TRIzol reagent (Ambion, Waltham, MA, USA) method. The extracted mRNA was subsequently reverse-transcribed (TaKaRa, Dalian, China) into cDNA. It is worth noting that hybrid poplar 84K (*Populus alba × Populus glandulosa*) has the same breeding environment as *‘Nanlin 95’* [64].

### 4.5. Quantitative Real-Time PCR Analysis

Each sample for qRT-PCR with TransStart^®^ Tip Green qPCR Super Mix (TransGen Biotech, Beijing, China) repeated at least three times on a CFX96 Real-Time System (Bio-Rad, Hercules, CA, USA). The thermocycler protocol was as follows: 94 °C for 30 s; 39 cycles of 94 °C for 5 s; and 60 °C for 30 s. The relative expression level was estimated using the 2^−∆∆Ct^ algorithm.

Primer Premier 5.0 software was used to design specific primers for each PtDi19 (Appendix A). The poplar *UBQ10* was as the reference gene, and studies have shown that *UBQ10* is the most stable reference gene under stress treatment [65]. The expression levels of ABA/drought-related genes in transgenic *Arabidopsis* were detected using the primers listed in Appendix A.

### 4.6. Subcellular Localization and Transactivation Activity

The full-length coding sequences without stop codon of *PtDi19-2* and *PtDi19-7* were successfully amplified and linked to pCAMBIA1305 vector containing CAMV35S and green fluorescent protein (GFP), respectively. The recombined vector was transformed into *Agrobacterium tumefaciens* EHA105. The bacterial fluid was then injected into tobacco leaves with a needle. The treated tobacco was instantaneously expressed in the dark for 36-40 h [66,67]. Finally, each sample was observed with the help of a confocal laser scanning microscope (CarlZeiss LSM710, Jena, Germany).

To investigate the transcriptional activity of PtDi19-2 and PtDi19-7, the CDS of *PtDi19-2, PtDi19-7*, and fragments of *PtDi19-2-N-C* and *PtDi19-2-N* were amplified and fused into the GAL4 DNA-binding domain of pGBKT7 (BD) vector. pGBKT7-53 + pGADT7-T were used as the positive experiment control, and empty plasmid (pGBKT7) was used as the negative control. We used lithium acetate method to treat the experimental groups PGBKT7-PtDi19-2, pGBKT7-PtDi19-2-N-C, pGBKT7-PtDi19-2-N, pGBKT7-PtDi19-7, pGBKT7, and pGBKT7-53+pGADT7-T transformed yeast cells, which were then inoculated on SD/-Trp and SD/-Trp/-His/-Ade/X-α-Gal and incubated in a constant temperature incubator at 30 ℃ for 4–5 days.

### 4.7. Yeast Two-Hybridization

PtDi19-2-N fragments were recombined into the pGBKT7 (BD) vector, and the full length of PtDi19-7 was recombined into the pGADT7 (AD) vector. The two recombinant plasmids (BD-PtDi19-2-N, AD-PtDi19-7) were co-transformed into yeast cells (AH109). The BD and AD-PtDi19-7 were co-transformed into yeast cells (AH109) as a negative control. The bacterial solutions of each group were diluted and inoculated on selective medium (SD/-Leu/-Trp and SD/-Leu/-Trp/-His/-Ade/X-α-Gal) [68]. The primer sequences are listed in Appendix A.

### 4.8. Phenotypic Analysis of Transgenic Plants

To study the effects of mannitol and ABA treatments on seed germination, *PtDi19-2* (OE2-3, OE2-4) and *PtDi19-7* (OE7-5, OE7-9) overexpression lines were cultured in 1/2MS, 1/2MS+ABA (0.7 µM), or 1/2MS+mannitol (300, 350, 400 mM) for 14 days, respectively. Each concentration corresponded to three sets of biological repeats, and each group contained 36 seeds. The photos were taken, and germination rate was counted. Additionally, GraphPad 8.3.0 software was used to visualize data in the form of histograms. Statistical analyses were performed and used to identify significant differences at * *p* <0.05 and ** *p* < 0.01 between WT and the 4 transgenic lines (OE2-3, OE2-4, OE7-5, and OE7-9) by one-way ANOVA in SPSS Statistics 17.0.

To determine the resistance of the transgenic lines, the 3-week-old plants (OE2-3, OE2-4, OE7-5, and OE7-9) had watering stopped for 10 days and were photographed. The relative water content, electrolyte leakage, proline content, malondialdehyde (MDA) content, and the peroxidase (POD) activity of wild-type (WT) plants and 4 transgenic lines were measured [69]. Using the method described in [70], the transgenic *Arabidopsis* leaves before and after drought treatment were taken and stained with diaminobenzidine (DAB). One-month-old wild-type 84 K poplar and *oxPtDi19-7* (ox#1, ox#2, and ox#3) transplanted seedlings were used as experimental materials for drought treatment. Phenotypic and physiological parameters were recorded and measured before and after drought treatment. The physiological parameters were determined by the same method as that of *Arabidopsis*. Roots of wild-type and transgenic lines treated with drought were taken out, and the length of roots was measured with a steel ruler, and the root weight of each line was weighed before and after drying.

### 4.9. ABA Sensitivity Test

One-month-old wild-type 84K poplar and *oxPtDi19-7* lines were selected as experimental materials. Stem segments of the same size, all containing a lateral bud, were inserted into 1/2 MS medium with or without 5 µM/10 µM ABA. Then, they were put into a greenhouse to grow for 21 days, and the results were observed and photographed.

The transgenic *Arabidopsis* leaves of the same location and similar size were immersed in (0 µM and 1 µM) ABA for 12 h, respectively. Then, the treated *Arabidopsis* leaves were put into a mixed solution containing 25% glycerol and chloral hydrate (2 g/mL) to remove chlorophyll and fix stomata. After 4 days, the stomata of the samples were observed under electron microscopy. The ratio of stomatal width to length is >0.5 (stomatal is open), 0.5–0.2 (stomatal is partially opened), and <0.2 (stomatal is closed) [71,72].

## 5. Conclusions

In conclusion, we identified eight PtDi19 genes in poplar. According to transcriptome data and qRT-PCR experiments, *PtDi19-2* and *PtDi19-7* were isolated. *PtDi19-2*-overexpressing and *PtDi19-7*-overexpressing lines improve drought tolerance and ABA sensitivity of transgenic *Arabidopsis*. In poplar, *oxPtDi19-7* could improve drought tolerance and ABA sensitivity of transgenic 84 K poplar. In addition, the overexpression *Arabidopsis* increased ABA-induced stomatal closure and the expression of ABA/drought-related genes. Therefore, this study suggested *PtDi19-2* and *PtDi19-7* could improve the drought tolerance of transgenic plants through ABA-dependent signaling pathways and provided a theoretical basis for poplar resistance molecular breeding.

## Figures and Tables

**Figure 1 ijms-23-03371-f001:**
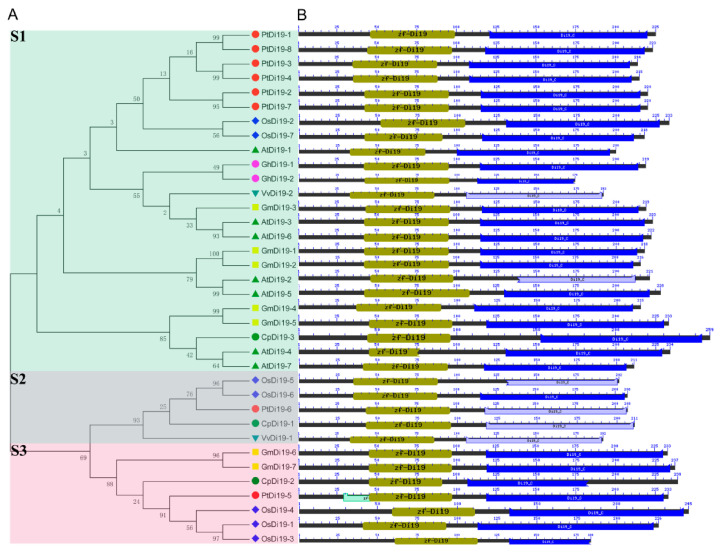
Phylogenetic tree and conserved protein structure of Di19s in seven species. (**A**) The phylogenetic tree constructed by the neighbor-joining method implemented by MEGA software. Different colors represent different subfamilies; Di19-S1/-S2/-S3 subfamilies use green, blue, and red, respectively. (**B**) Conserved protein domain analysis of Di19s. The amino acid length, zf-Di19 domain, and Di19_C domain are represented by black, olive green, and blue, respectively. The length of each pattern is displayed proportionally.

**Figure 2 ijms-23-03371-f002:**
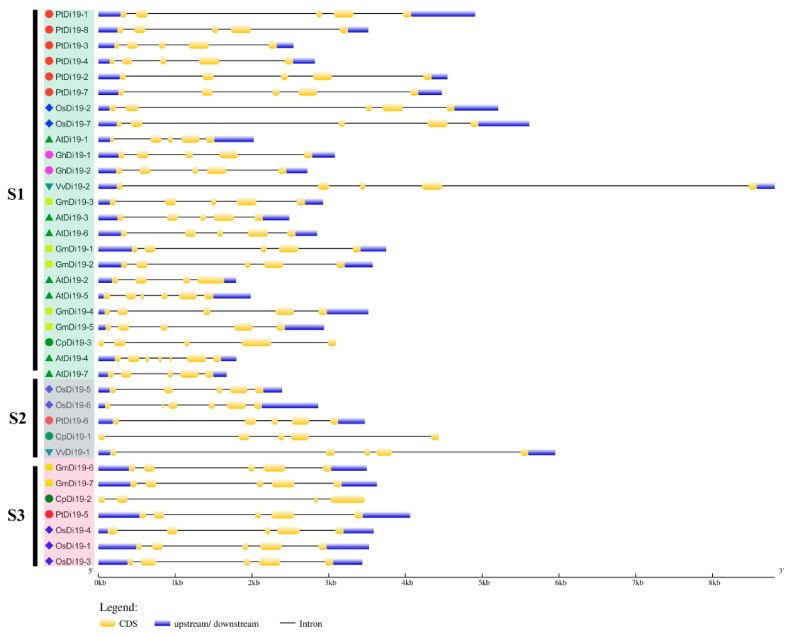
Gene structure of Di19s in seven species. Use GSDS online tool for gene structure analysis. Yellow boxes and black lines represent exons and introns, respectively. Blue box indicates the 5′ and 3′ non-coding regions. The length represents the size of exon and intron.

**Figure 3 ijms-23-03371-f003:**
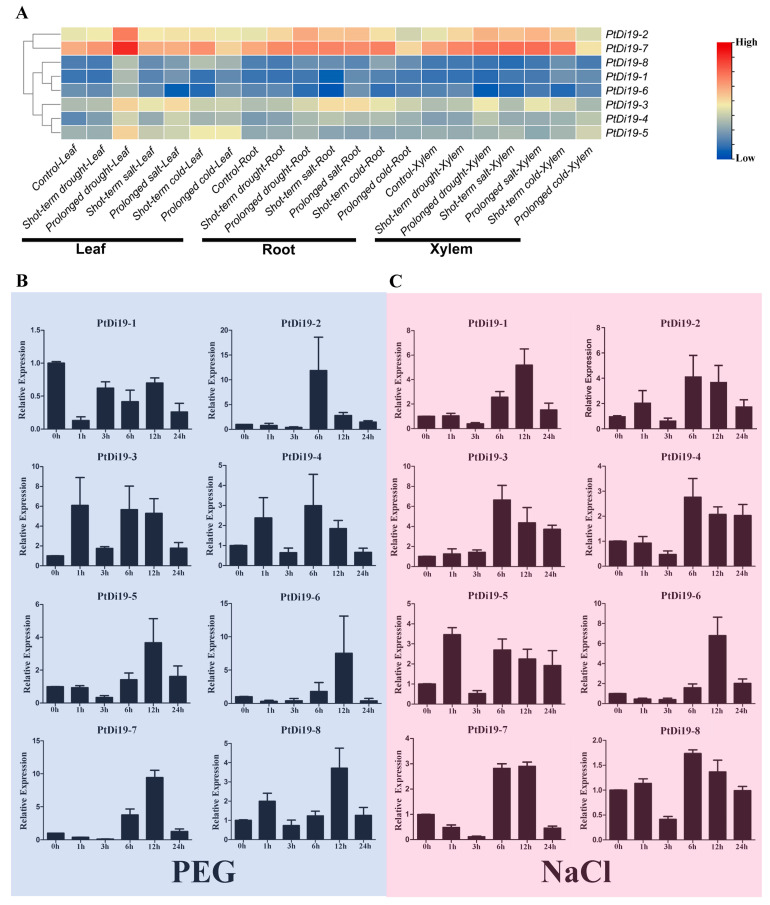
RNA-seq and qRT-PCR analysis of PtDi19 genes under stress. (**A**) Heat maps of the expression levels of eight PtDi19 genes in different tissues under drought, salt, and cold treatment. The legend was to show the relative high and low expressions, value = log_2_ (fold change). (**B**,**C**) The induced expression pattern of PtDi19 with 20% PEG 6000 and 200 mmol/L NaCl root irrigation, respectively. The *Y*-axis indicates the relative expression levels, and 0 h, 1 h, 3 h, 6 h, 12 h, and 24 h (*X*-axis) indicate hours of treatment.

**Figure 4 ijms-23-03371-f004:**
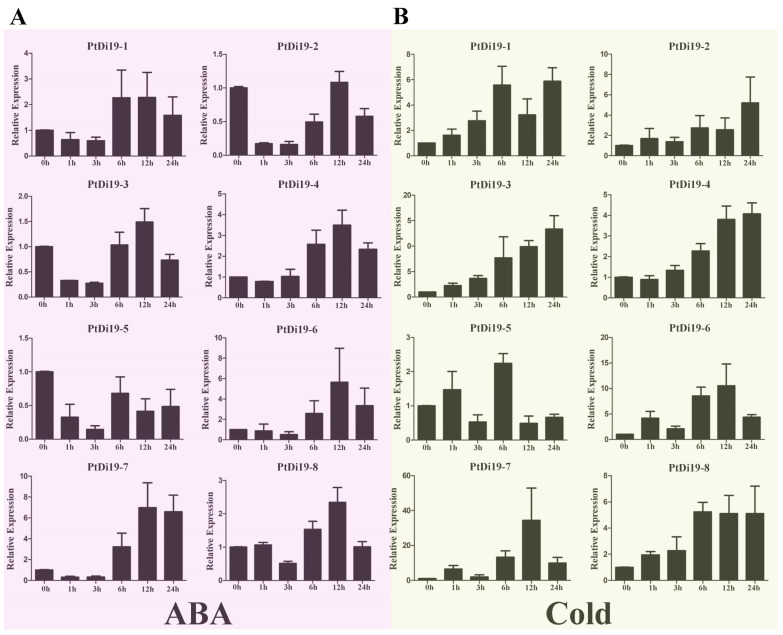
qRT-PCR analysis of the expression profile of PtDi19 gene. (**A**,**B**) The induced expression pattern of PtDi19 with 100 µmol/L ABA root irrigation and 4 °C low temperature treatment, respectively. The *Y*-axis indicates the relative expression levels, and 0 h, 1 h, 3 h, 6 h, 12 h, and 24 h (*X*-axis) indicate hours of treatment.

**Figure 5 ijms-23-03371-f005:**
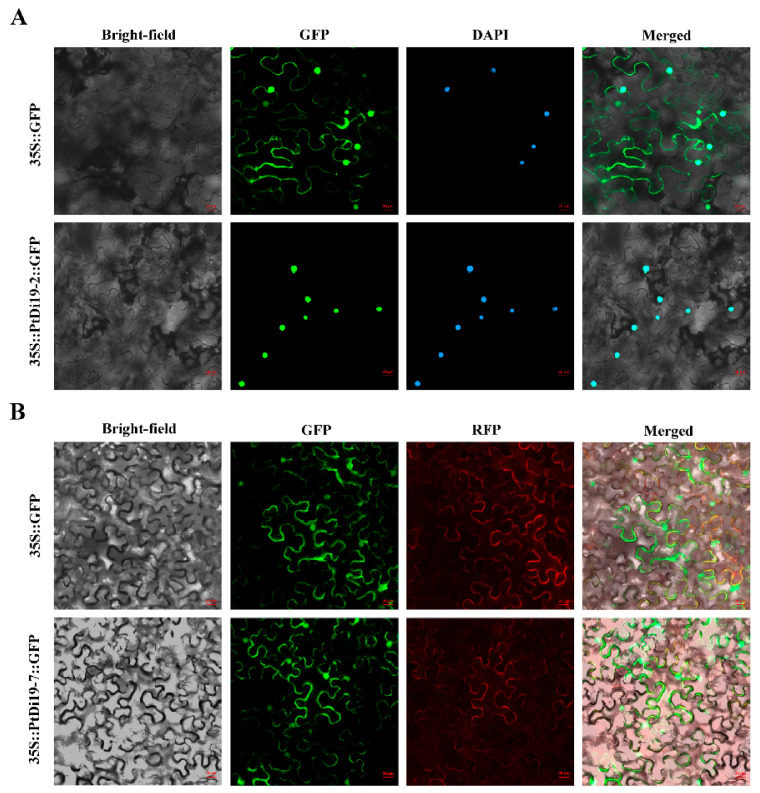
Subcellular localization of PtDi19–2 and PtDi19–7 in tobacco leaves. (**A**) The control and 35S::PtDi19–2::GFP fusion protein was separately expressed in tobacco leaves and observed by the use of a fluorescence microscope. The signal of the GFP channel exhibits a green color, and DAPI staining for the nucleus presents as a blue fluorescence; the bright field was jointly used for forming the merged channel. (**B**) The control and 35S::PtDi19–7::GFP fusion protein were separately expressed in tobacco leaves and observed by the use of a fluorescence microscope. The signal of the GFP channel exhibits a green color, and RFP channel exhibits red fluorescence; the bright field was jointly used for forming the merged channel. Scale bars = 20 µm.

**Figure 6 ijms-23-03371-f006:**
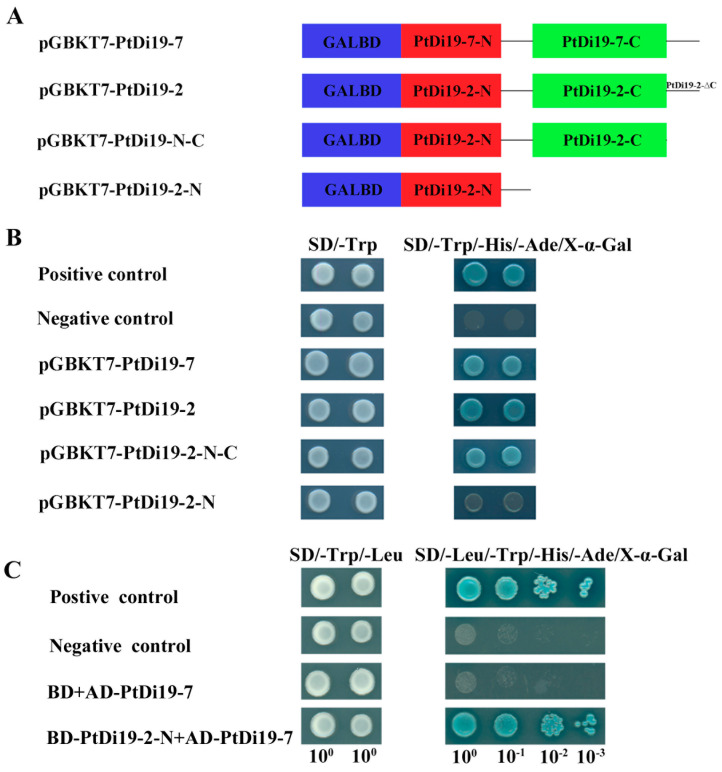
PtDi19–2 and PtDi19–7 proteins both had transcriptional activity and interaction with each other in yeast. (**A**) A schematic diagram illustrating the PtDi19–7 and cDNA fragments encoding different portions of PtDi19–2, which were fused to DNA sequences encoding the GAL DNA binding domain in the yeast vector pGBKT7. (**B**) Transactivation activity of the PtDi19–2 and PtDi19–7 protein in yeast. (**C**) Recombinant plasmids of BD–PtDi19–2–N and AD–PtDi19–7 co-transformed into yeast strain AH109, and then plated on a selective medium.

**Figure 7 ijms-23-03371-f007:**
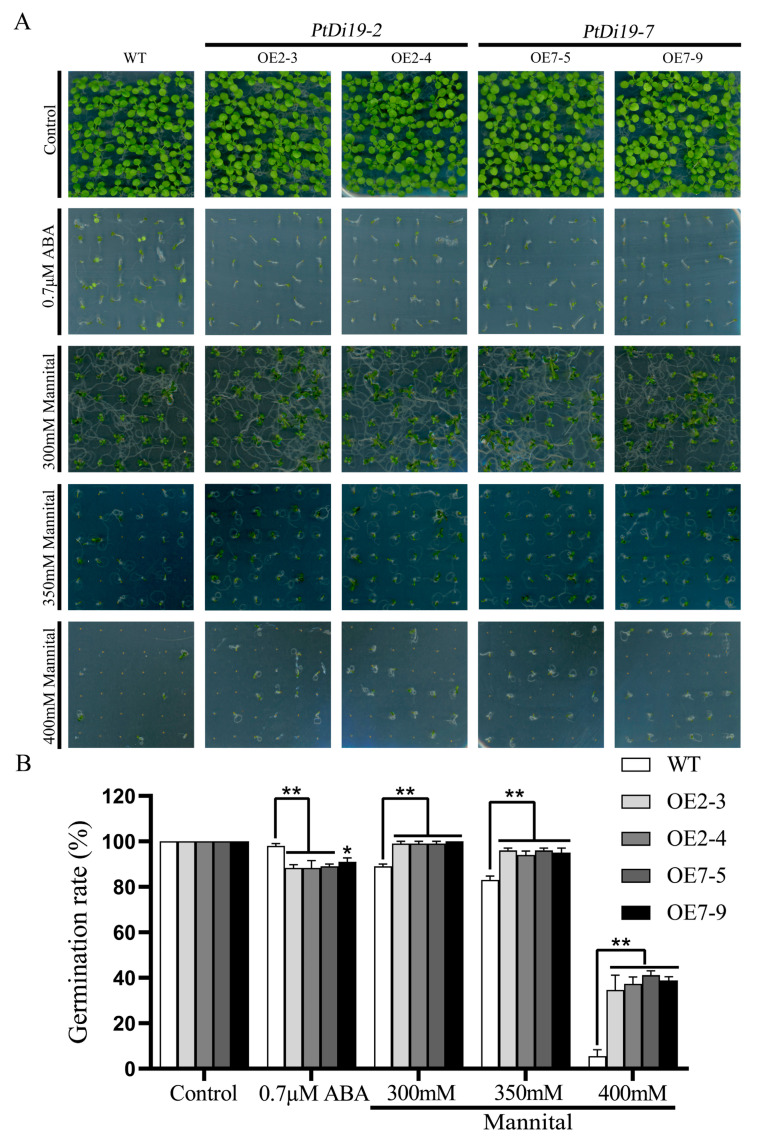
The overexpression *Arabidopsis* of *PtDi19**–2* and *PtDi19**–7* were sensitive to ABA and showed strong drought tolerance. (**A**) We spotted the seeds of WT and overexpression lines evenly on 1/2MS, 1/2MS+0.7 µMABA, or 1/2MS+mannitol (300 mM, 350 mM, and 400 mM) to observe seed germination. (**B**) Counting of the germination rate of different lines. Asterisks indicate a significant difference compared to the corresponding controls (* *p* < 0.05 and ** *p* < 0.01). Error bars indicate SEs from three replicates.

**Figure 8 ijms-23-03371-f008:**
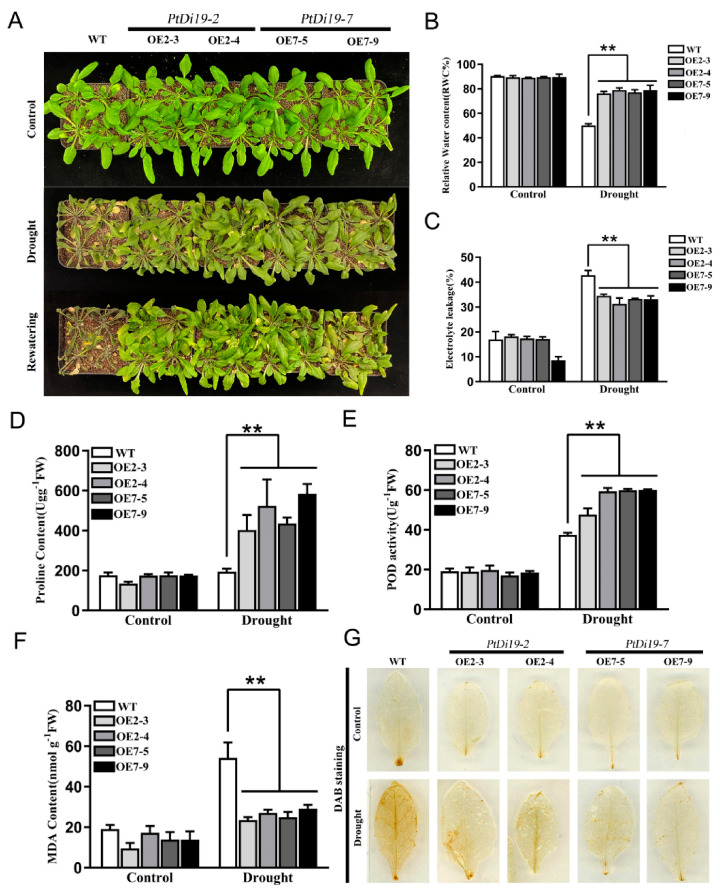
The *PtDi19**–2* and *PtDi19**–7* improved drought resistance in *Arabidopsis*. (**A**) Three week WT and transgenic plants were withheld water for 10 days to induce dehydration. After dehydration for 10 days, the representative images taken. (**B**) Relative water content of the leaves. (**C**) Electrolyte leakage. (**D**) Proline contents. (**E**) POD activity. (**F**) MDA content. (**G**) DAB staining. A *p*-value of <0.01 was considered to be extremely significant (**).

**Figure 9 ijms-23-03371-f009:**
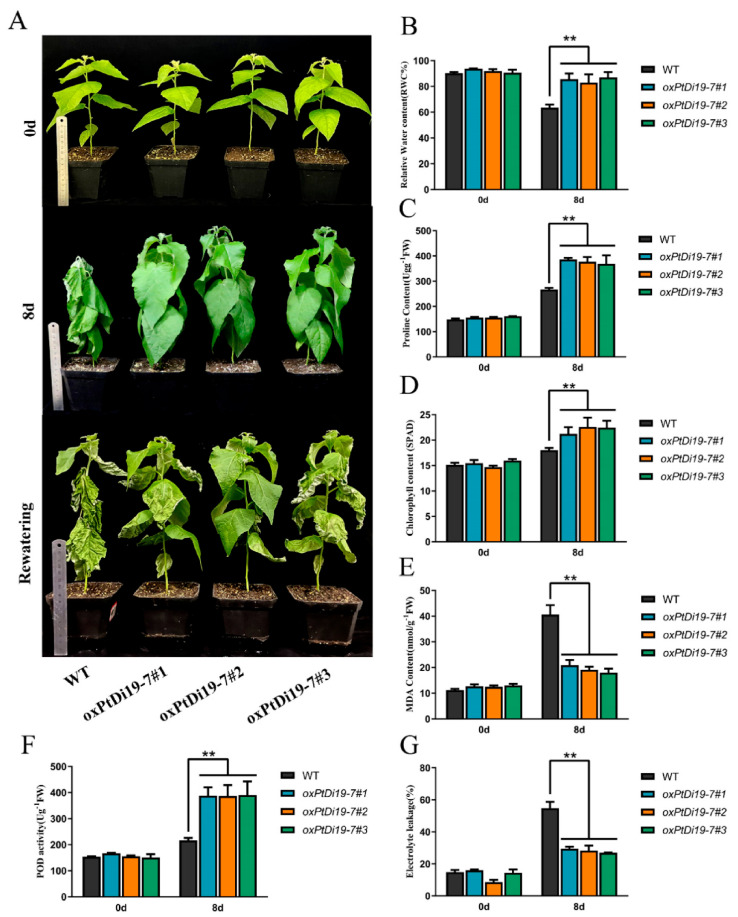
*PtDi19**–7* improved drought resistance in poplar. (**A**) One-month-old wild-type 84K poplar and *oxPtDi19**–7* lines were withheld water for 8 days to induce dehydration. After dehydration for 8 days, the representative images were taken. (**B**) Relative water content of the leaves. (**C**) Proline contents. (**D**) Chlorophyll content. (**E**) MDA content. (**F**) POD activity. (**G**) Electrolyte leakage. A *p*-value of <0.01 was considered to be extremely significant (**).

**Figure 10 ijms-23-03371-f010:**
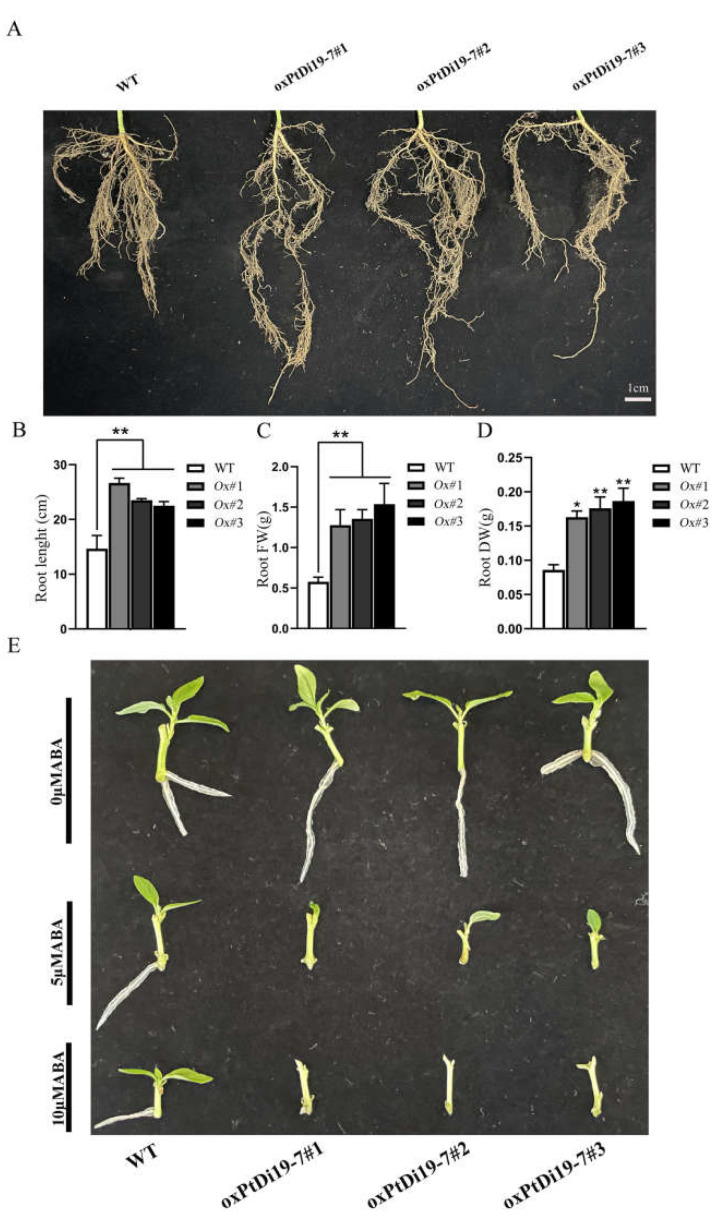
*PtDi19**–7* affects root development of transgenic poplar and responds to exogenous ABA. (**A**) Phenotypic analysis of root length after 8 days of drought treatment. (**B**) The root length measurement. (**C**) The root fresh weight. (**D**) The root dry weight. (**E**) Lateral bud outgrowth of short shoot segments grown for 3 weeks on 1/2 MS medium supplemented with ABA (5/10 µM) or without ABA of WT and *oxPtDi19**–7* plants. A *p*-value of <0.05 was considered to be significant (*), and a *p*-value of <0.01 was considered to be extremely significant (**).

## Data Availability

The datasets generated during and/or analyzed during the current study are available from the corresponding author on reasonable request.

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
