# Peer review of "Homologous Drought-Induced 19 Proteins, PtDi19-2 and PtDi19-7, Enhance Drought Tolerance in Transgenic Plants"

_ijms, 2022, doi:10.3390/ijms23063371_

Round 1
Reviewer 1 Report
The manuscript entitled "A pair of homologous Drought-induced 19 proteins, PtDi19-2 2 and PtDi19-7, enhance drought tolerance in transgenic plants" is very well written and the authors reported the Di19 proteins first time in poplar.
i have very minor queries before final acceptence of the paper
- authors should added some latest refrences
- how the Chromosome mapping has been done should be littlebit more elaborated.
- the full name of PEG should be given in Abbreviation
Reviewer 2 Report
Title - Authors can remove "a pair" from the title. The term "Homologs" reflect the same meaning.
Authors need to significantly improve the abstract. The flow of the information is haphazard.
"The PtDi19-2 and PtDi19-7, with high expression under drought stress, both have transcriptional activity. " -this does not make any sense
Introduction
The use of "pair" is confusing, it can confuse with tandemly duplicated gene pair.
"involved in the positive regulation of ABA-dependent drought." here not clear what author means by ABA-dependent drought.
Materials and methods
"Eight PtDi19 genes were identified and distributed separately on chromosomes" - looks inappropriate, need to rephrase.
Results
"The largest protein was PtDi19-5 (233 amino acids) and the smallest protein was PtDi19-6 (208 amino acids). The molecular weight of the proteins ranged from 22.9 kDa (PtDi19-6) to 26.3 kDa (PtDi19-5), and the pI ranged from 4.44 (PtDi19-1) to 6.23 (PtDi19-6)." - Authors can avoid such description and only provide descriptive data in the supplementary tables.
"Among 36 Di19 genes, 20 conserved motifs were identified as motifs 1 to 20" - here authors got 20 motifs because they set the limit 20, if they change the limit of motifs in MEME the number will change. Therefore I suggest using background normalization to identify motifs specific to Di19.
Better to move figure 2 to supplementary. The prediction of the cis-regulatory element has very less reliability due to the high level of random occurrence. Therefore I suggest condensing the section.
In figure 7, except 40mm concentration rest, all treatment seems to have a non-significant difference. How authors have analysed the replications. I am curious to know how many seeds per replication and replications per treatment were considered.
Reviewer 3 Report
Dear authors
The current manuscript entitled “A pair of homologous Drought-induced 19 proteins, PtDi19-2 and PtDi19-7, enhance drought tolerance in transgenic plants” demonstrated the role of PtDi19-2 and PtDi19-7 positively regulate the drought stress through ABA-dependent signaling pathways. The present manuscript is very well organized and presented. Please find below some queries for further improvement.
- Line 23- please explain what does 84K in poplar means?
- Line 52-78- This information suits more as a discussion part, rather than introduction.
- Line 101 says “Eight PtDi19 genes were identified and distributed separately on chromosomes 2, 5, 8, 10, 11, 12, 13 and 14.” Please reframe this sentence.
- Line 498- 502- what is the significance of cold stress after 20% PEG-6000 solution, 200 mM saturated sodium chloride (NaCl) solution and 100 µM (ABA)? Is it a combination of stress, please clear.
- Section 4.4.- does it denotes the transgenic plants?
- Only one reference gene (poplar UBQ10) has been used?
- Section 4.8- please specify which transgenic lines.
Thank you
